# Muscular Strength and Carotid Intima–Media Thickness in Physically Fit Young Adults: The CHIEF Atherosclerosis Study

**DOI:** 10.3390/jcm11185462

**Published:** 2022-09-16

**Authors:** Gen-Min Lin, Kun-Zhe Tsai, Yun-Chen Chang, Wei-Chun Huang, Xuemei Sui, Carl J. Lavie

**Affiliations:** 1Department of Internal Medicine, Hualien Armed Forces General Hospital, Hualien City 970, Taiwan; 2Department of Internal Medicine, Tri-Service General Hospital, National Defense Medical Center, Taipei 114, Taiwan; 3Department of Stomatology of Periodontology, Mackay Memorial Hospital, Taipei 104, Taiwan; 4School of Nursing and Graduate Institute of Nursing, China Medical University, Taichung 406, Taiwan; 5College of Medicine, National Yang Ming Chiao Tung University, Yangming Campus, Taipei 112, Taiwan; 6Department of Critical Care Medicine, Kaohsiung Veterans General Hospital, Kaohsiung 813414, Taiwan; 7Department of Exercise Science, Arnold School of Public Health, University of South Carolina, Columbia, SC 29208, USA; 8John Ochsner Heart and Vascular Institute, Ochsner Clinical School, The University of Queensland School of Medicine, New Orleans, LA 70121, USA

**Keywords:** carotid intima–media thickness, muscular strength, push-ups, sit-ups, young adults

## Abstract

**Background:** Greater muscular strength (MusS) has been found to have an inverse association with subclinical atherosclerosis in children, as well as with mortality from cardiovascular diseases (CVDs) in middle-aged and elderly individuals. However, the association of the degree of MusS with atherosclerosis may differ by sex and has not been clarified in young adults. **Methods and Results:** A total of 1021 Taiwanese military personnel, aged 18–40 years, participated in annual health examinations in 2018–2020. MusS was separately assessed by 2-min push-up and 2-min sit-up numbers. Subclinical atherosclerosis was measured by the left carotid bulb intima–media thickness (cIMT) using high-resolution ultrasonography. Multiple linear regression with adjustments for age, sex, alcohol intake, cigarette smoking, anthropometric indices, blood pressure, and lipid profiles was utilized to determine the correlation between MusS and cIMT. Both 2-min push-up and 2-min sit-up numbers were inversely correlated with cIMT (standardized β: −0.089 and −0.072, respectively; both *p*-values < 0.05). In men, both 2-min push-up and 2-min sit-up numbers were inversely correlated with cIMT (standardized β: −0.076 and −0.086, respectively; both *p*-values < 0.05), while in women, 2-min push-up numbers but not 2-min sit-up numbers were inversely correlated with cIMT (standardized β: −0.204 and −0.01; *p* = 0.03 and 0.99, respectively). **Conclusions:** Among young adults, there was an inverse association between MusS and cIMT, emphasizing the beneficial impact of MusS on the regression of atherosclerosis. The study also revealed a sex difference and suggested that training of the upper arm muscles may be an effective preventive measure for young women to reduce the risk of early cardiovascular diseases.

## 1. Introduction

Atherosclerosis is the main cause of clinical cardiovascular diseases (CVDs), e.g., coronary heart disease, peripheral artery occlusive disease, and ischemic stroke [1]. The progression of atherosclerosis is slow and starts in early life [2]. Several traditional risk factors of atherosclerosis have been identified, such as smoking, dyslipidemia, hypertension, obesity, and physical inactivity [2,3,4,5]. Current evidence shows that CVD events could be greatly reduced by taking preventive measures for these modifiable risk factors [6,7,8]. It is important to know the leading causes of subclinical atherosclerosis at young ages [9] so that preventive measures can be taken early for reducing the progression of atherosclerosis and the development of clinical CVD.

Carotid intima–media thickness (cIMT)—a marker of subclinical atherosclerosis, measured by high-resolution B-mode ultrasonography [9,10,11]—has been significantly associated with first-time premature CVD events [9]. In the youth and young adults, factors associated with cIMT include obesity, blood pressure (BP), dyslipidemia, and physical fitness [12]. In terms of physical fitness, cardiorespiratory fitness (CRF) has been reported to have an inverse association with cIMT in young individuals [11,12,13]. There have only been a few studies on the associations between muscular strength (MusS) and cIMT in children [10] and premature CVD death in adolescents [14], despite the fact that several studies have found associations between MusS and diabetes mellitus, CVD death, sudden cardiac death, and heart failure in middle-aged or elderly individuals [15,16,17,18,19,20,21]. Notably, the association could vary by sex, and was dependent on which MusS was assessed [22]. In addition, many studies were not adjusted for CRF levels to clarify the independent role of MusS in the prevention of atherosclerosis [14,15,22]. Today, the association of MusS with cIMT in young adults remains unclear and has not been previously studied. It is crucial to provide more evidence to fill the gap in understanding of the association between MusS and cIMT in young adults. Therefore, this study aimed to investigate the association between MusS and cIMT in a military population of physically fit young men and women in Taiwan.

## 2. Materials and Methods

### 2.1. Study Population

The cardiorespiratory fitness and health in eastern armed forces study (CHIEF) [23,24,25] for the evaluation of subclinical atherosclerosis included 1822 military male and female participants from Taiwan, aged 18–40 years, without the use of any anti-hypertensive or lipid-lowering agents, from 2018 to 2020. All participants received daily exercise training at a military base in the morning for longer than 6 months, including a time-limited 3000-m run (in 25 min) for endurance capacity training, and 20 successive push-ups and 20 successive sit-ups or more within 10 min following the run for MusS training. Each participant underwent annual health examinations for laboratory and physical status screening [26], and self-reported a questionnaire regarding toxic substance use habits—e.g., cigarette smoking and alcohol consumption (active versus former or never)—in the Hualien Armed Forces General Hospital. Each participant also received high-resolution B-mode ultrasonography over the left carotid artery bulb for the measurement of cIMT. The exclusion criteria in the study were those who could not complete the MusS tests (the 2-min push-up and 2-min sit-up tests) and the endurance capacity test (the 3000-m run test) consecutively in the same day of the annual military exercise testing, which was performed after the health examinations and before the end of the index year. 

### 2.2. Physical and Blood Laboratory Examinations 

The BP of each participant was measured once over right arm in a sitting position, after a rest for at least 15 min, using the same automatic monitor device (FT201, Parama-Tech Co., Ltd., Fukuoka, Japan), via the oscillometric method. Anthropometric parameters including waist circumference (WC), body height, and body weight were measured in a standing position. Body mass index (BMI) was defined as the ratio of body weight (kg) to the square of body height (m^2^). The blood laboratory tests for total cholesterol, high-density lipoprotein (HDL-C), triglycerides, uric acid, and fasting glucose were measured using an auto-analyzer (AU640, Olympus, Kobe, Japan). Each blood sample was obtained by certificated medical technologists following an overnight 12-h fast [27,28,29,30,31]. 

### 2.3. The Measurements for cIMT 

The cIMT was quantitatively measured from the leading edge of the media–adventitia interface to the leading edge of the lumen–intima interface of the far wall of the left carotid bulb, using an ultrasound scanner equipped with a linear 4–8 MHz probe on the iE33 machine (Philips Medical Systems, Andover, MA, USA). The coefficient of variation for the repeated assessments of the cIMT of the left carotid bulb was estimated to be 96.8%. In addition, there were no carotid plaques detected during the sonographic procedure for any of the participants [13].

### 2.4. MusS and CRF Assessments

Each participant’s muscular strength was evaluated by 2-min push-up and 2-min sit-up numbers. The 2-min limited time for brief bursts of push-up and sit-up exercises was based on the findings of previous studies [32,33]. Each strength test was performed once in the main building of the Military Physical Training and Testing Center from 14:00 to 15:00, and the order could be interchangeable. Both exercise procedures were performed on sponge pads and scored using electronic or infrared sensors. In the sit-up test, the examinees’ hands were attached to their ears and their feet were fixed by anchors on the sponge pad. The examinees scored if their upper trunk bended forward and their elbows hit the electronic sensors on both thighs. In the push-up test, the examinees scored with the movement of elbow flexion and extension when their head, back, and buttocks could keep a line with the initial level set at the priming time, as detected by the infrared sensors. The push-up test was prematurely ceased if body parts other than the hands and toes touched the sponge pad before the time ran out. 

The CRF of each participant was assessed by time for a 3000-m run test. The run test was carried out outdoors on a flat playground in the Military Physical Training and Testing Center at 16:00, following the MusS tests after a rest for one hour. The run test was permitted only if the coefficient of the heat stroke risk formula—defined as the product of outdoor temperature on the Celsius scale and relative humidity (%) × 0.1—was lower than 40 and there was no heavy rain. 

### 2.5. Statistical Analysis 

Clinical characteristics of the participants are presented as means ± standard deviations for continuous variables and numbers (percentage) for categorical variables. Univariate linear regression was used to determine the individual correlations of 2-min push-up numbers, 2-min sit-up numbers (MusS), and 3000-m run time (CRF) with cIMT. Multiple linear regression analysis was separately utilized to determine the associations of the strength of various muscles and CRF with cIMT, with adjustments for age, sex, cigarette smoking, alcohol intake, BMI, WC, systolic BP, diastolic BP, serum triglycerides, total cholesterol, and HDL-C (Model 1). The associations of 2-min sit-up and 2-min push-up numbers with cIMT were further adjusted for CRF in Models 2 and 3, respectively. Scatterplots were drawn between 2-min push-up numbers, 2-min sit-up numbers, and 3000-m run time. Subgroup analysis for each exercise test was performed by sex (men and women). In addition, multiple logistic regression analysis was separately utilized to determine the odds ratios (ORs) of CRF and MusS levels stratified by ±1 standard deviation (16%, 68%, and 16%) for cIMT ≥ 0.90 mm—a cut-off value considered to be clinically significant [13]. The initial covariates adjusted in Model 1a for multiple logistic regression included age, sex, cigarette smoking, and alcohol intake, followed by Model 1b, which was the same as the first multiple linear regression. A value of *p* < 0.05 was regarded as statistically significant. All analyses were carried out using SPSS version 25.0 for Windows (IBM Corp., Armonk, NY, USA). This study has been reviewed and approved by the Institutional Review Board (IRB) of the Clinical Ethics Committee of the Mennonite Christian Hospital (No. 16-05-008) in Hualien City, Eastern Taiwan, R.O.C., and written informed consent was obtained from all participants. 

## 3. Results

### 3.1. Clinical Characteristics of the Participants

Of the total participants, those with body mass index (BMI) ≥ 30 kg/m^2^ (N = 274) or serum triglycerides ≥ 400 mg/dL (N = 28) who were not allowed to attend the annual exercise testing, and those who did not complete the three exercise tests in the same day due to bad weather outdoors preventing the run test (N = 499), were excluded from this study, leaving a final sample of 1021 participants for analysis. The clinical characteristics of the study population are shown in Table 1. The participants’ mean age was 27.6 years, and their mean BMI was 24.6 kg/m^2^. The majority of the participants were male, accounting for 89.3%. In total, there were 408 (40.0%) active smokers and 372 (36.5%) active alcohol consumers. The mean time (in seconds) for a 3000-m run test was 893.6, the mean 2-min sit-up numbers were 46.1, and the mean 2-min push-up numbers were 45.2. The mean cIMT was estimated to be 0.70 mm, and 42 participants (4.1%) were found to have cIMT ≥ 0.9 mm.

### 3.2. Correlations between MusS, CRF Levels, and cIMT among All Participants

Table 2 shows the univariate and multiple linear regression results of MusS and CRF with cIMT among all participants. In the univariate linear analysis, MusS separately assessed by 2-min push-up and 2-min sit-up numbers, along with CRF evaluated 3000-m run time, was inversely correlated with cIMT (standardized β: −0.089, −0.072, and 0.114, respectively; all *p* < 0.05). In the first multiple linear regression, the findings were consistent with the univariate linear regression results (standardized β: −0.099, −0.083, and 0.119, respectively; all *p* < 0.05). While CRF was intentionally put in Model 1 to assess the independent association of each MusS with cIMT, the inverse correlation was significant for 2-min push-up numbers in Model 3 (standardized β: −0.080, *p* = 0.02) and was marginal for 2-min sit-up numbers in Model 2 (standardized β: −0.062, *p* = 0.08). In Models 2 and 3, CRF was inversely and significantly correlated with cIMT (standardized β: 0.107 and 0.104, respectively; both *p* < 0.05). In Figure 1, the scatterplots reveal the results for the correlation of CRF with the 2-min push-up numbers and 2-min sit-up numbers (R = 0.377 and 0.370, respectively; both *p* < 0.001), as well as the correlation between 2-min push-up numbers and 2-min sit-up numbers (R = 0.762, *p* < 0.001). With regard to the other covariates for cIMT, age was directly correlated with cIMT in a crude model, Model 2, and Model 3 (standardized β: 0.062, 0.073, and 0.072, respectively; all *p* = 0.04). The direct correlation for triglycerides was significant in Model 1 (standardized β: 0.078, *p* = 0.04) and was borderline significant in Models 2 and 3 (standardized β: 0.072 and 0.070, respectively; both *p* = 0.07).

### 3.3. Correlations of MusS and CRF Levels with cIMT in Men and Women

The results of univariate and multiple linear regression for the correlation of MusS and CRF with cIMT in men and women are shown in Table 3. In a crude model for men, CRF was inversely correlated with cIMT (standardized β: 0.100, *p* = 0.003), whereas the inverse correlations for 2-min sit-up numbers and 2-min push-up numbers were borderline (standardized β: −0.060 and −0.057, respectively; *p* = 0.07 and 0.08, respectively). Conversely, for women, there were no correlations for CRF and 2-min sit-up numbers, except that there was a marginal inverse correlation for 2-min push-up numbers with cIMT (standardized β: −0.175, *p* = 0.07) in univariate analysis. In the multiple linear regression analysis for men (Model 1), CRF, 2-min sit-up numbers, and 2-min push-up numbers were inversely correlated with cIMT (standardized β: 0.111, −0.086, and −0.076, respectively; all *p* < 0.05). For women, similar to the univariate analysis results, 2-min push-up numbers were inversely and significantly correlated with cIMT (standardized β: −0.204, *p* = 0.03), whereas there were no correlations for CRF and 2-min sit-up numbers. 

### 3.4. Associations of MusS and CRF Levels with Significant cIMT ≥ 0.90 mm 

In Table 4, in both the run test and the 2-min sit-up test, as compared with those in the top 16% of exercise performance, the risks of cIMT ≥ 0.90 mm were not significant in participants whose exercise performance was in the middle 68% or the bottom 16% in Models 1a and 1. On the other hand, there was a trend for a greater risk of cIMT ≥ 0.90 mm with lower performance in the 2-min push-up test in Models 1a and 1b (*p* for trend = 0.03 and 0.049, respectively). In addition, as compared with those in the top 16% for push-up performance, participants in the bottom 16% for push-up performance had a greater risk of cIMT ≥ 0.90 mm in Model 1a (OR: 4.09 (95% confidence interval (CI): 1.03–16.33); *p* = 0.04), and the association was borderline in Model 1b (OR: 3.86 (95% CI: 0.96–15.64); *p* = 0.06).

## 4. Discussion

The principal finding of this study was that in physically active young adults, MusS was inversely correlated with cIMT. In line with the findings of previous studies, the association of some MusS—e.g., for the upper arms with cIMT—was independent of CRF, while for other MusS—e.g., the lower thighs and psoas muscles—the association may have been related to CRF in this study. In addition, the association for various types of MusS may differ by sex, as both 2-min push-up and 2-min sit-up numbers were correlated with cIMT in men, while only 2-min push-up numbers were correlated with cIMT in women. We also found that age and serum triglycerides were independently correlated with cIMT, as has been well demonstrated in previous studies [9,13].

Several studies have demonstrated better CRF to be a protector against subclinical atherosclerosis, CVD events, and related death in the general population across a wide age range [10,11,12,13,14]. However, there have been relatively fewer studies with regard to the associations between MusS, atherosclerosis severity, and the risk of CVD—particularly among the youth and young adults [11,14]. Melo et al. showed that MusS assessed by handgrip strength was inversely correlated with cIMT (independently of CRF) in children [11]. Ortega et al. revealed an inverse association between MusS—separately evaluated by handgrip strength and knee extension—and the occurrence of CVD mortality in adolescents. However, the association of MusS of the upper arms—evaluated by elbow flexion—with CVD mortality was not confirmed [14]. In the study of Katzmarzyk and Craig, 68% of the participants were aged younger than 40 years, and an inverse association of MusS assessed by 1-min sit-up numbers with all-cause mortality was observed. On the other hand, the association between grip strength and unlimited-time push-up numbers was null in their study participants [22]. A summary of these studies is provided in Table 4. The present study found an inverse association of 2-min push-ups with cIMT, which was not affected by CRF and was greater than that of 2-min sit-ups. Our findings were not consistent with the reports of Ortega et al. [14] or Katzmarzyk and Craig [22], where the strength of the thighs assessed by knee extension or sit-ups—rather than the strength of upper arms assessed by elbow flexion or push-ups—independently predicted CVD and all-cause mortality. This heterogeneity may be due to the differences in age distribution, ethnicity, physical training, and outcomes of interest between the previous studies [14,22] and the present study.

In addition, previous studies also showed a sex difference in the association of MusS with CVD and all cause-mortality [15,22]. Gale et al. [19] found that in a sample of elderly individuals, grip strength was inversely associated with all-cause mortality in men, but not in women. Katzmarzyk and Craig [22] also demonstrated the same findings, as the model was adjusted for age only in a population mainly composed of young adults (Table 4) [22]. In the present study, we found a sex difference in the association of sit-ups and CRF with cIMT, which was significant in men but not in women. The mechanism for the inverse association of MusS with mortality and cIMT in men may be partially due to the impact of greater MusS in reducing insulin resistance—a vital contributor to atherosclerosis—evidenced in young adults and older individuals [34,35]. For the inconsistent findings in women, Rantanen et al. [36] found handgrip strength to be a powerful predictor of mortality from CVD and from all causes in moderately-to-severely disabled old women. It is obvious that the baseline functional status and physical activity of the population studied might affect the results for the relationship of MusS with atherosclerosis and mortality risk. In summary, in a fragile population, those who have greater physical fitness obtain more benefits from reducing their risk of CVD. Another possible mechanism for the sex difference may be that, in general, men have higher levels of physical activity and CRF than women [27,37], possibly leading to a stronger association of MusS—assessed by various strength tests—with cIMT and mortality.

The data in the present study certainly suggest that efforts should be made to increase MusS throughout society. Although there may be a genetic component to MusS, the best way to increase MusS is through physical activity and, in particular, resistance exercise (RE) [7,21]. Indeed, substantial evidence exists to show that RE improves CVD risk factors and reduces the risk of death from CVD, even independent of aerobic exercise and CRF [7,21,38,39,40].

### Study Strengths and Limitations

The major strength of this study was that the unrecognized confounders were minimized by the similar lifestyle and training program followed by participants from the military. Second, the health examinations, along with CRF and MusS testing, were standardized using the same protocols. Third, the present study included a large number of participants, providing sufficient power to perform the subgroup analysis by sex, in which the statistical power for men and women was estimated to be 1.00 and 0.80, respectively, despite the fact that women made up only 10.7% of the overall population. The different results achieved in men and women may be due to the smaller number of women, and should not downplay this difference. On the other hand, there were some limitations in the present study. First, because this was a cross-sectional study, the temporal changes in MusS assessed by 2-min sit-ups and 2-min push-ups with cIMT could not be investigated. Second, the cIMT was only measured over the left carotid bulb of each participant, and there may be a modest difference between the left and right cIMT, possibly leading to a minor variation in the results. Finally, this study lacked multiethnic/racial diversity, making the generalization of the results difficult.

## 5. Conclusions

This study’s findings revealed a sex difference in the association of various MusS testing with cIMT, and suggested that training of the upper arm muscles may be an effective preventive measure for young women to reduce the risk of early CVD. In addition, this study reconfirmed that among physically active young adults there was an inverse association between MusS and cIMT, emphasizing the beneficial effect of greater MusS on atherosclerosis. 

## Figures and Tables

**Figure 1 jcm-11-05462-f001:**
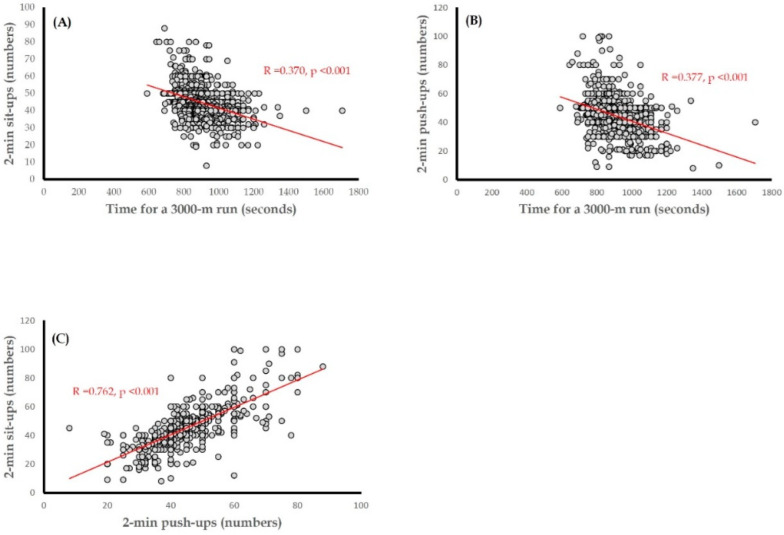
Scatterplots for the correlation between (**A**) 2-min sit-up numbers and time for a 3000-m run test, (**B**) 2-min push-up numbers and time for a 3000-m run test, and (**C**) 2-min sit-up numbers and 2-min push-up numbers.

**Table 1 jcm-11-05462-t001:** Clinical characteristics of the participants.

	N = 1021
Age, years	27.61 ± 5.87
Sex, male (%)	911 (89.3)
Smoking, active (%)	408 (40.0)
Alcohol intake, active (%)	372 (36.5)
Body mass index, kg/m^2^	24.59 ± 3.54
Waist circumference, cm	82.31 ± 9.55
Systolic blood pressure, mmHg	116.97 ± 12.83
Diastolic blood pressure, mmHg	69.09 ± 9.88
Total cholesterol, mg/dL	173.64 ± 33.16
Low-density lipoprotein, mg/dL	107.08 ± 30.40
High-density lipoprotein, mg/dL	50.75 ± 10.96
Serum triglycerides, mg/dL	104.03 ± 80.49
Fasting plasma glucose, mg/dL	93.62 ± 12.76
Serum uric acid, mg/dL	6.48 ± 1.45
Time for a 3000-m run, sec	893.55 ± 106.19
2-min sit-ups, numbers	46.05 ± 9.34
2-min push-ups, numbers	45.20 ± 11.71
cIMT ≥ 0.9 mm	42 (4.1)

Abbreviations: cIMT, left carotid bulb intima–media thickness.

**Table 2 jcm-11-05462-t002:** Correlations between muscular strength, cardiorespiratory fitness, and cIMT of all participants.

	Crude Model			Model 1			Model 2			Model 3		
	R	β*	*p*	R	β*	*p*	R	β*	*p*	R	β*	*p*
Time for a 3000-m run	0.11	0.114	<0.001	0.17	0.119	0.001	0.18	0.107	0.003	0.18	0.104	0.004
2-min sit-ups numbers	0.07	−0.072	0.02	0.15	−0.083	0.02	0.18	−0.062	0.08			
2-min push-ups numbers	0.09	−0.089	0.004	0.16	−0.099	0.006				0.18	−0.080	0.02
Age	0.06	0.062	0.04	0.13	0.056	0.11	0.18	0.073	0.04	0.18	0.072	0.04
Male Sex	0.06	0.062	0.04	0.13	0.043	0.22	0.18	−0.035	0.40	0.18	−0.044	0.28
Active smoking	0.05	−0.045	0.15	0.13	−0.053	0.13	0.18	−0.059	0.09	0.18	−0.058	0.10
Active alcohol intake	0.01	−0.012	0.69	0.13	0.016	0.64	0.18	0.027	0.44	0.18	0.027	0.43
Body mass index	0.01	0.011	0.73	0.13	0.096	0.13	0.18	0.098	0.13	0.18	0.103	0.11
Wasit circumference	0.03	−0.027	0.39	0.13	−0.072	0.28	0.18	−0.102	0.13	0.18	−0.110	0.10
Systolic blood pressure	0.03	−0.027	0.38	0.13	0.009	0.84	0.18	0.028	0.52	0.18	0.028	0.52
Diastolic blood pressure	0.05	−0.054	0.08	0.13	−0.052	0.23	0.18	−0.062	0.15	0.18	−0.062	0.15
Total cholesterol	0.02	−0.024	0.43	0.13	−0.040	0.28	0.18	−0.034	0.36	0.18	−0.030	0.41
High density lipoprotein	0.03	0.025	0.43	0.13	0.028	0.45	0.18	0.032	0.40	0.18	0.031	0.42
Serum triglycerides	0.02	0.021	0.49	0.13	0.078	0.04	0.18	0.072	0.07	0.18	0.070	0.07

Multiple linear regression analysis was used to determine the associations between muscular strength and cIMT, with the following adjustments—Model 1: age, sex, smoking, alcohol intake, body mass index, waist circumference, systolic blood pressure, diastolic blood pressure, total cholesterol, high-density lipoprotein, and serum triglycerides; Model 2: age, sex, smoking, alcohol intake, body mass index, waist circumference, systolic blood pressure, diastolic blood pressure, total cholesterol, high-density lipoprotein, serum triglycerides, 3000 m run time, and 2-min sit-up numbers; Model 3: age, sex, smoking, alcohol intake, body mass index, waist circumference, systolic blood pressure, diastolic blood pressure, total cholesterol, high-density lipoprotein, serum triglycerides, 3000 m run time, and 2-min push-up numbers; β* denotes standardized β. Abbreviations: cIMT, left carotid bulb intima–media thickness.

**Table 3 jcm-11-05462-t003:** Correlations between muscular strength, **cardiorespiratory fitness**, and cIMT between men and women.

	Crude Model						Model 1					
	Men			Women			Men			Women		
	R	β*	*p*	R	β*	*p*	R	β*	*p*	R	β*	*p*
3000-m run time	0.10	0.100	0.003	0.08	0.080	0.40	0.17	0.111	0.001	0.37	0.119	0.23
2-min sit-up numbers	0.06	−0.060	0.07	0.02	0.019	0.84	0.15	−0.086	0.01	0.35	−0.001	0.99
2-min push-up numbers	0.06	−0.057	0.08	0.18	−0.175	0.07	0.15	−0.076	0.02	0.41	−0.204	0.03

Multiple linear regression analysis was used to determine the correlations between muscular strength and cIMT, with the following adjustments for Model 1: age, sex, smoking, alcohol intake, body mass index, waist circumference, systolic blood pressure, diastolic blood pressure, total cholesterol, high-density lipoprotein, and serum triglycerides; β* denotes standardized β. Abbreviations: cIMT, left carotid bulb intima–media thickness.

**Table 4 jcm-11-05462-t004:** Associations of mscular strength and cardiorespiratory fitness levels with cIMT ≥ 0.9 mm in all participants.

	Model 1a		Model 1b	
	OR (95% CI)	*p*	OR (95% CI)	*p*
CRF (time for a 3000 m run)				
Top 16% performance level	1.00		1.00	
Middle 68% performance level	1.40 (0.53–3.67)	0.49	1.44 (0.55–3.79)	0.46
Bottom 16% performance level	1.65 (0.53–5.16)	0.38	1.70 (0.54–5.35)	0.36
*p*-Value for trend		0.36		0.51
MusS (sit-up numbers)				
Top 16% performance level	1.00		1.00	
Middle 68% performance level	1.66 (0.57–4.82)	0.35	1.57 (0.54–4.61)	0.40
Bottom 16% performance level	2.24 (0.61–8.25)	0.22	2.13 (0.58–7.86)	0.25
*p*-Value for trend		0.22		0.25
MusS (push-up numbers)				
Top 16% performance level	1.00		1.00	
Middle 68% performance level	2.41 (0.73–8.01)	0.15	2.33 (0.70–7.79)	0.17
Bottom 16% performance level	4.09 (1.03–16.33)	0.04	3.86 (0.96–15.64)	0.06
*p*-Value for trend		0.03		0.049

Multiple logistic regression analysis (Model 1a) was used to determine the associations between physical fitness and cIMT, with adjustments for age, sex, smoking, and alcohol intake. Model 1 was used with adjustments for age, sex, smoking, alcohol intake, body mass index, waist circumference, systolic blood pressure, diastolic blood pressure, total cholesterol, high-density lipoprotein, and serum triglycerides. Abbreviations: cIMT, left carotid bulb intima–media thickness; CRF, cardiorespiratory fitness; MusS, muscular strength; OR, odds ratio; CI, confidence interval.

## Data Availability

The datasets generated and/or analyzed during this study are not publicly available due to materials obtained from the military in Taiwan, which are confidential, but are available from the corresponding author upon reasonable request.

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
