# Peer review of "Muscular Strength and Carotid Intima–Media Thickness in Physically Fit Young Adults: The CHIEF Atherosclerosis Study"

_jcm, 2022, doi:10.3390/jcm11185462_

Round 1

Reviewer 1 Report

The authors present the results of a study where they compared muscular strength and carotid intima-media thickness inphysically active young adults. Among the she strengths of their cohort, of note is the high number of patients included.

However, there are also some concerns about this study:

-       Each strength test was performed once, and no training for the test is mentioned, which could lead to an underperformance of some participants because of insufficient understanding of how to perform the tests

-       Lack of multi-ethnic/racial diversity, making the generalization of the results difficult; can do the authors comment on this?

-        Significantly fewer women were included and associations between muscular strengths and cIMT were mostly significant only for men. I wonder how reliable the results in this group are, considering the high number of patients needed for statistical power. Were power calculations performed?

-       The mean time (secs) for a 3000-meter run test was 893.6, which is equivalent to almost 15 minutes, a better than the average time for this distance, suggesting experienced runners were included. In addition, the mean cIMT reported in this study was estimated 0.70 mm, which is a normal value. Do the authors consider these comparisons clinically meaningful? 

-       Since these participants underwent annual health examinations, including for physical status, were there any follow-ups performed in them? Were there cardiovascular events registered if follow-ups existed?

-       The authors mention that multivariable models including age, sex, smoking, drinking, BMI, waist circumference, SBP, DBP, total cholesterol, HDL- 168 C and triglycerides were performed, but the results for these predictors are not presented, nor discussed. These results are important and should be reported in the article

-       The authors conclude that there is a beneficial effect of muscular strength on atherosclerosis regression, although they previously report that is a cross-sectional study, so progression/regression cannot be measured. In addition, the authors did not report in the results section the existence of atherosclerosis in these participants. This should be clarified.

Author Response

Journal: Journal of Clinical Medicine

Ref: jcm-1859166 R1

Title: Muscular Strength and Carotid Intima-Media Thickness in Physically Active Young Adults: CHIEF Atherosclerosis Study

Dear Editor:

Thank you very much for the opportunity to revise our manuscript. We have revised our paper using point-by-point response to the reviewers’ comments and highlighted in red as follow.

 Reviewer Comments:

Reviewer 1

Comments and Suggestions for Authors

The authors present the results of a study where they compared muscular strength and carotid intima-media thickness in physically active young adults. Among the strengths of their cohort, of note is the high number of patients included.

Response:

Thank you very much for your kindly comment.

However, there are also some concerns about this study:

-       Each strength test was performed once, and no training for the test is mentioned, which could lead to an underperformance of some participants because of insufficient understanding of how to perform the tests

Response:

Thank you very much for this important comment.

We have added statements regarding the training for the strength tests and cardiorespiratory fitness test on lines 37-41, page 5, in the revised manuscript as you suggested.

“All participants received daily exercise training for longer than 6 months at military base in the morning, including a time-limited 3000-meter run (in 25 minutes) for endurance capacity training, and 20 successive push-ups and 20 successive sit-ups or more within 10 minutes following the run for MusS training.”

-       Lack of multi-ethnic/racial diversity, making the generalization of the results difficult; can do the authors comment on this?

Response:

Thank you very much for this important comment.

We have added this point in our limitation section on lines 272-274, page 13, in the revised manuscript, as you suggested.

-        Significantly fewer women were included and associations between muscular strengths and cIMT were mostly significant only for men. I wonder how reliable the results in this group are, considering the high number of patients needed for statistical power. Were power calculations performed?

Response:

The statistical power for men and women have been added on lines 263-267, page 12 in the revised manuscript, as you suggested.

“Third, the present study included a large number of participants, providing sufficient power to perform the subgroup by sex in which the statistical power for men and women was estimated 1.00 and 0.80, respectively, despite that, women were merely 10.7% of the overall population.”

-       The mean time (secs) for a 3000-meter run test was 893.6, which is equivalent to almost 15 minutes, a better than the average time for this distance, suggesting experienced runners were included. In addition, the mean cIMT reported in this study was estimated 0.70 mm, which is a normal value. Do the authors consider these comparisons clinically meaningful? 

Response:

Since our study participants were physically fit military personnel, the exercise performances and cIMT were more likely robust.

To respond to your comment, we have added an analysis for the association of muscular strength with significant cIMT which was defined as ≥0.9 mm in Table 4. The results were also described on lines 182-192, page 10.

Associations of MusS and CRF Levels with Significant cIMT ≥0.90 mm

In Table 4, in both of the run test and the 2-minute sit-up test, as compared with those with top 16% exercise performance, the risks of significant cIMT were not significant in participants whose exercise perofrmance was in middle 68% and bottom 16% in models 1a and 1. On the contrary, there was a trend for a greater risk of significant cIMT with lower performances in the 2-minute push-up test in model 1a and model 1 (p for trend =0.03 and 0.049). In addition, as compared with those with top 16% push-ups performance, participants with bottom 16% push-ups performance had a greater risk of significant cIMT in model 1a [OR: 4.09 (95% confidence interval (CI): 1.03-16.33), p =0.04] and the association was borderline in model 1 (OR: 3.86 (95% CI: 0.96-15.64), p =0.06).”

-       Since these participants underwent annual health examinations, including for physical status, were there any follow-ups performed in them? Were there cardiovascular events registered if follow-ups existed?

Response:

The measurement of cIMT was not routinely performed for the military personnel and thus we could not compare the interval change in the cIMT. In addition, since the participants were included in 2018-2020, there were no cardiovascular events registered in the follow-up till now.

-       The authors mention that multivariable models including age, sex, smoking, drinking, BMI, waist circumference, SBP, DBP, total cholesterol, HDL-C and triglycerides were performed, but the results for these predictors are not presented, nor discussed. These results are important and should be reported in the article

Response:

Thank you very much for this important comment.

The results for the other covariates have been provided on lines 158-162, page 9, and the discussion was briefly made on lines 202-204, page 10, as they were not the main goals of the present study.

On lines 158-162, page 9:

“With regard to the other covariates for cIMT, age was correlated with cIMT in a crude model, model 2 and model 3 (standardized β: 0.062, 0.073 and 0.072, all p =0.04). The correlation for serum triglycerides was significant in model 1 (standardized β: 0.078, p =0.04) and was borderline significant in models 2 and 3 (standardized β: 0.072 and 0.070, both p =0.07).”

On lines 202-204, page 10:

“We also revealed that age and serum triglycerides were independently correlated with cIMT, which have been well demonstrated in the previous studies.”

-       The authors conclude that there is a beneficial effect of muscular strength on atherosclerosis regression, although they previously report that is a cross-sectional study, so progression/regression cannot be measured. In addition, the authors did not report in the results section the existence of atherosclerosis in these participants. This should be clarified.

Response:

Thank you very much for this important comment.

The conclusion of the text has been rewritten as you suggested.

“This study revealed a sex difference in the association of various MusS testing with cIMT, and suggests that upper arm muscles training may be an effective preventive measure for young women to reduce the risk of early CVD. In addition, this study reconfirmed that among physically active young adults, there was an inverse association between MusS and cIMT, emphasizing the beneficial effect of greater MusS on atherosclerosis.”

In addition, information for the existence of atherosclerosis in participants have been provided on lines 71-72, page 6 in the revised manuscript as you suggested.

“In addition, there were no carotid plagues detected during the sonographic procedure for all participants.”

Reviewer 2 Report

Comments to Authors:

I would like to congratulate the authors for the study, but the paper should be reviewed by a native English speaker. 

  • The background should rewrite me again. The authors do not explain the text well, and therefore the text is not true.
  • The introduction is very short. Authors should better explain and justify the text.
  • The authors should remove table 4 from the discussion. The table does not provide additional information.
  • The authors must rewrite the conclusion. The main objective of the study is considered as an additional conclusion. The second conclusion must be the main one.
  • The style of reference is not correct.

Author Response

Journal: Journal of Clinical Medicine

Ref: jcm-1859166 R1

Title: Muscular Strength and Carotid Intima-Media Thickness in Physically Active Young Adults: CHIEF Atherosclerosis Study

Dear Editor:

Thank you very much for the opportunity to revise our manuscript. We have revised our paper using point-by-point response to the reviewers’ comments and highlighted in red as follow.

Reviewer Comments:

Reviewer 2

Comments and Suggestions for Authors

I would like to congratulate the authors for the study, but the paper should be reviewed by a native English speaker. 

Response:

Thank you very much for your kindly comment.

The paper has been reviewed by Drs. Carl J. Lavie and Xuemei Sui, who are the co-authors of the paper and both are native English speaker.

  • The background should rewrite again. The authors do not explain the text well, and therefore the text is not true.

Response:

Thank you very much for this important comment.

The background of the abstract has been rewritten as you suggested.

“Greater muscular strength (MusS) has been found with an inverse association with subclinical atherosclerosis in children, and cardiovascular diseases (CVD) mortality in middle- and old-aged individuals. However, the association of various MusS with atherosclerosis may differ by sex and has not been clarified in young adults.”

  • The introduction is very short. Authors should better explain and justify the text.

Response:

Thank you very much for your kindly comment.

The introduction content has been enriched to provide better explains and justify the text in the revised manuscript as you suggested.

  • The authors should remove table 4 from the discussion. The table does not provide additional information.

Response:

Table 4 has been removed from the discussion section as you suggested.

  • The authors must rewrite the conclusion. The main objective of the study is considered as an additional conclusion. The second conclusion must be the main one.

Response:

Thank you very much for this important comment.

The conclusion of the text has been rewritten as you suggested.

“This study revealed a sex difference in the association of various MusS testing with cIMT, and suggests that upper arm muscles training may be an effective preventive measure for young women to reduce the risk of early CVD. In addition, this study reconfirmed that among physically active young adults, there was an inverse association between MusS and cIMT, emphasizing the beneficial effect of greater MusS on atherosclerosis.”

  • The style of reference is not correct.

Response:

We have corrected the references according to the Journal’s style as you suggested.

Round 2

Reviewer 2 Report

congratulations to the authors, they have answered my comments satisfactorily

Author Response

Dear Reviewer,

Thank you very much for your review, making our paper more scientific.

Best regards.

Gen-Min Lin, MD, PhD, FACC, FAHA

Chief, Hualien Armed Forces General Hospital, Taiwan